# Surveillance and Genetic Analysis of Low-Pathogenicity Avian Influenza Viruses Isolated from Feces of Wild Birds in Mongolia, 2021 to 2023

**DOI:** 10.3390/ani14071105

**Published:** 2024-04-04

**Authors:** Yong-Myung Kang, Erdene-Ochir Tseren Ochir, Gyeong-Beom Heo, Se-Hee An, Hwanseok Jeong, Urankhaich Dondog, Temuulen Myagmarsuren, Youn-Jeong Lee, Kwang-Nyeong Lee

**Affiliations:** 1Animal and Plant Quarantine Agency, 177 Hyeoksin 8-ro, Gimcheon-si 39660, Republic of Korea; kym34@korea.kr (Y.-M.K.); imheo@korea.kr (G.-B.H.); ashpri@korea.kr (S.-H.A.); hs2369@korea.kr (H.J.); leeyj700@korea.kr (Y.-J.L.); 2Department of Infectious Disease and Microbiology, School of Veterinary Medicine, Mongolian University of Life Sciences, Zaisan Street, Ulaanbaatar 17024, Mongolia; erkavet@muls.edu.mn (E.-O.T.O.); urankhaich.d@muls.edu.mn (U.D.); 1019099@muls.edu.mn (T.M.)

**Keywords:** avian influenza, surveillance, feces, wild bird, Mongolia

## Abstract

**Simple Summary:**

The introduction of novel highly pathogenic avian influenza (HPAI) viruses into Korea has been linked to recombination events at breeding sites in the Northern Hemisphere. This has prompted increased monitoring and genetic analysis of avian influenza viruses (AIVs) in northern regions like Mongolia, sharing migratory bird flyways with Korea. From 2021 to 2023, 10,149 fecal samples from wild birds were analyzed in Mongolia, revealing a 1.01% AI virus prevalence with 77 isolated strains, including various *HA* and *NA* subtypes. Although HPAI viruses were not detected, genetic analysis showed close relations between Mongolian and Korean AI viruses, emphasizing ongoing surveillance due to the potential introduction of HPAI strains from Mongolia.

**Abstract:**

The introduction of novel highly pathogenic (HPAI) viruses into Korea has been attributed to recombination events occurring at breeding sites in the Northern Hemisphere. This has increased interest in monitoring and genetically analyzing avian influenza viruses (AIVs) in northern regions, such as Mongolia, which share migratory bird flyways with Korea. AIVs in Mongolia were monitored by analyzing 10,149 fecal samples freshly collected from wild birds from April to October in 2021 to 2023. The prevalence of AIVs in wild birds was 1.01%, with a total of 77 AIVs isolated during these 3 years. These 77 AIVs included hemagglutinin (*HA*) subtypes H1, H2, H3, H4, H6, H10 and H13 and neuraminidase (*NA*) subtypes N1, N2, N3, N6, N7 and N8. The most frequently detected subtype combinations were H3N8 (39.0%) and H4N6 (19.5%), although HPAI viruses were not detected. Genetic analysis indicated that theses AIVs isolated from Mongolian samples were closely related to AIVs in wild birds in Korea, including those of Eurasian lineage. These findings indicate the necessity of continuous AIV surveillance and monitoring, as HPAI viruses introduced into Korea may derive from strains in Mongolia.

## 1. Introduction

Avian influenza virus (AIV), a type A influenza virus, has been found to induce highly contagious and lethal respiratory diseases in wild birds as well as in gallinaceous poultry, depending on strains [1]. This RNA virus has been classified into different subtypes based on the specific types of two envelope proteins, hemagglutinin (*HA*) and neuraminidase (*NA*) [2,3]. These subtypes, which are crucial for viral infection and transmission, are highly variable and may be responsible for interspecies transmission and adaptation [2]. For this, AIVs evolve primarily through viral mutations and the reassortment of viral genomic segments originating from various strains [4,5]. Given that such variant viruses can be transmitted over long distances through wild birds, AI surveillance for neighboring countries is necessary [6].

The emergence of novel strains of highly pathogenic AIVs (HPAI viruses) has been attributed to viral mutation and rearrangement occurring at breeding and overwintering sites located in the Northern Hemisphere [7,8]. The presence of these newly emergent strains in wild birds underscores the potential for the dissemination of these viruses along migratory bird flyways [9]. In Mongolia, approximately 476 species of wild birds have been recorded, spanning across 60 families and 19 orders. Among them, 81 species are resident birds, while 395 species are migratory birds [10]. Rich and distinct communities of wild birds are present in Mongolia, a large territory encompassing various natural zones and landscapes [11]. These habitats serve as gathering and breeding grounds for numerous migratory birds that travel over two primary flyways: the East Asian-Australasian flyway, which includes both Korea and Mongolia, and the Central Asian flyway [12].

In Korea, HPAI viruses, especially H5Nx viruses from various clades of the Goose/Guangdong lineage, have been seasonally detected in poultry since 2003 [13]. It has been suggested that viral reassortment occurs at breeding sites in Siberia, and that these newly generated variants are introduced into East Asia, including Korea, during the winter migration of wild birds [14]. Many species of migratory birds from Siberia and neighboring regions, including Swan Geese (*Anser cygnoides*) and Mute Swans (*Cygnus olor*), transit through Mongolia on their way to East Asia during the winter season [15]. In East Asia, there is a pressing need for comprehensive research addressing both AI surveillance and migration of wild birds in Mongolia. Research groups in Korea and Japan have therefore collaborated with those in Mongolia in the active surveillance of wild birds for the earlier detection of newly emergent reassorted AIVs [16,17,18].

From 2021 to 2023, a collaboration between Korean researchers and researchers at the Mongolian University of Life Sciences (MULS) has monitored fecal samples from wild birds in selected Mongolian habitats to detect and isolate AIVs. The present study describes the isolation and genomic sequences of these viruses, as well as reporting the results of molecular epidemiological analysis to understand the relevance of these AIVs to South Korea.

## 2. Materials and Methods

### 2.1. Sample Collection

Fresh fecal samples were collected from wild birds in three distinct ecogeographic regions of Mongolia, the eastern, central and western regions, between April and October from 2021 to 2023. In the western region, samples were primarily collected at Khar-Us Lake and Uvs Lake. In the central region, samples were collected at Khunt Lake, Erkhel Lake, Sharga Lake, Tuin River and Doitiin Tsgaaan Lake. In the eastern region, samples were collected from Chukh Lake, Gurem Lake and Ganga Lake (Figure 1 and Appendix A). These ecogeographic regions, including Khunt Lake and Doitiin Tsgaaan Lake [19], were selected because they are key wild bird habitats and locations of previous HPAI outbreaks in wild birds (Table 1). Fresh fecal samples were collected from areas near waterfronts where waterfowls were observed gathering or roosting. All fecal samples were transported to the diagnostic laboratory of MULS under refrigerated conditions at 4 °C.

### 2.2. Molecular Detection and Virus Isolation

All fecal samples were suspended in 1% gentamycin PBS buffer, vortexed and centrifuged at 3500 rpm for 5 min. Total RNA was extracted from 200 µL of each supernatant using a Viral Gene-spin Viral DNA/RNA Extraction kit (iNtRON Biotechnology, Seong-Nam, Republic of Korea). The extracted RNAs were screened via real-time reverse transcription polymerase chain reaction (rRT-PCR) using Quantitect probe RT-PCR kits (Qiagen, MD, USA) on a StepOnePlus Real-Time PCR system (Thermo Fisher Scientific, MA, USA) to detect the Matrix (*M*) gene. Samples showing threshold cycle (Ct) values under 40 were defined as positive for the *M* gene [20].

Samples positive for the *M* gene were inoculated into 10-day-old embryonated chicken eggs, which were incubated for 72 h at 37 °C. Allantoic fluid was harvested from egg cultures and subjected to hemagglutination assays (HAs).

### 2.3. Molecular and Phylogenetic Analysis

Viral RNA was extracted from *HA*-positive allantoic fluids using NX-48 Viral NA kits (Genolution, Seoul, Republic of Korea) following the manufacturer’s instructions. All eight genome segments of the isolates were amplified by RT-PCR [21]. Viruses were subjected to complete genome sequencing using next-generation sequencing with the Illumina MiSeq platform, employing the Nextera DNA Flex Library Prep Kit (Illumina, San Diego, CA, USA) in accordance with the manufacturer’s instructions. Genomic sequences were analyzed by CLC Genomics Workbench 23 (Qiagen, Valencia, CA, USA). The nucleotide sequences of the viruses isolated in this study have been submitted to the GISAID database under accession numbers EPI_ISL_18968070, EPI_ISL_18968158, EPI_ISL_18968191, EPI_ISL_18968192, EPI_ISL_18968238–EPI_ISL_18968261, EPI_ISL_18968552–18968553 and EPI_ISL_18968661. The reference datasets for all gene segments subjected to the phylogenetic analysis were downloaded from the Genbank and GISAID EpiFlu databases. Nucleotide phylogenetic trees were built using the RAxML maximum-likelihood (RaxML) and MAFFT on CIPRES Science Gateway [22]. The reliability of the trees was assessed via bootstrap analysis with 1000 replicates. The tree displays were generated by the interactive Tree of Life (iTOL) program [23].

## 3. Results and Discussion

South Korea has experienced continual outbreaks of HPAI since 2003, and since 2020, these outbreaks have caused significant socio-economic losses during four consecutive winter seasons (2020–2024) [13]. Previous studies suggested that the incursion of HPAI viruses into East Asia is predominantly attributed to viral strains originating from breeding grounds in Eurasia [15]. So, for early warning and preemptive disease-control measures, comprehensive monitoring of circulating AIVs at breeding sites located in northern countries, such as Mongolia, would be necessary. In this regard, researchers in Mongolia and Korea collaborated in AIV surveillance from 2006 to 2009 [17]. The present study describes the results of a new collaboration in AIV surveillance, beginning in 2021, between researchers at the Animal and Plant Quarantine Agency (APQA) in Korea and the MULS in Mongolia.

A total of 10,149 fecal samples were collected from wild birds in western, central and eastern Mongolia from 2021 to 2023 (Table 1), including 4416 collected in 2021, 2137 in 2022, and 3866 in 2023. Of the 4146 samples collected in 2021, 39 (0.94%) were positive for the *M* gene. Isolation and sequencing of 30 viruses out of 39 *M*-positive samples showed that 19 were of subtype H3N8, 7 were of subtype H4N6 and 4 were of subtype H3N3. Of the 2137 samples collected in 2022, 33 (1.54%) were positive for *M*-positive samples. Viruses were isolated from 24 of the 30 *M*-positive samples, with subsequent sequencing showing that 5 were of subtype H3N8, 4 each were of subtypes H4N6 and H6N1, 2 each were of subtypes H2N8, H3N2, H3N3 and H10N7, and 1 each was of subtypes H1N2, H3N1 and H6N8. Of the 3866 samples collected in 2023, 31 (0.80%) were positive for *M* gene viruses isolated from 23 of the 31 *M*-positive samples, with subsequent sequencing showing that 6 were of subtype H3N8, 4 were of subtype H4N6, 3 each were of subtypes H3N2, H4N1 and H10N8, 2 were of subtype H6N1, and 1 each was of subtypes H1N2 and H13N8. Among the 77 virus isolates, some were found to be mixed with more than two AIVs in the assemblies against the reference sequences of H1 to H16 and N1 to N9 subtypes; in these cases, the subtypes within a higher coverage of depth were chosen for classification. Further study is needed to conduct additional analyses on quasispecies of AIV.

Over the 3-year period, the prevalence of AIV in wild birds in Mongolia was relatively low (1.01%), and the locational prevalence of AIV was highest (1.5%) at Khar-Us (47°50′32.4″ N, 92°01′33.2″ E) Lake in western Mongolia (Appendix A). This low prevalence was consistent with results showing that only 1.37% and 1.7% of samples were from wild birds in Mongolia [17,18]. The average prevalence of AIVs in samples from wild birds in China was slightly higher at 2.5%, with higher rates in Central China and among *Anseriformes* [24]. These findings suggested that targeted active surveillance, including the capture of wild birds and collection of feces, has potential for the early detection of AIVs in wild birds [25]. Moreover, targeted surveillance and identification of risk factors may be essential for effective AIV prevention and control. However, it is also crucial to elevate the viral detection rate in fecal samples. Thus, continuous research, such as the present study, is needed to investigate optimal sampling sites, timing and efficient transportation methods for samples, especially for countries like Mongolia.

The 77 AIVs sequenced included viruses of *HA* subtypes H1, H2, H3, H4, H6, H10 and H13, and *NA* subtypes N1, N2, N3, N6, N7 and N8. H3 (54.5%) and H4 (23.4%) were the most abundant *HA* subtypes, followed by H6 (9.1%), H10 (6.5%), two each for H1 (2.6%) and H2 (2.6%), and one for H13 (1.3%). The most frequently detected *NA* subtype was N8 (48.1%), followed by N6 (19.5), N1 (13.0%), N3 (7.8%) and N2 (9.1%). The most frequently detected subtype combinations were H3N8 (39.0%) and H4N6 (19.5%), followed by H3N3, H6N1, H3N2, H4N1, H10N8, H1N2, H2N8, H10N7, H3N1, H6N8 and H13N8 (Table 2). These findings support the outcomes of the surveillance studies of AIVs in wild birds in Mongolia, conducted in 2009–2013, 2016–2018 and 2017–2019 [17,18]. It is therefore likely that AIVs have intermingled across various migratory bird species along intersecting flyways, facilitating the long-distance spread of the virus within Asia and across other continents [26].

Phylogenetic analysis of the surface genes of isolated Mongolian AIVs was performed to assess their genetic relationships with other AIVs, including those in Korea (Figure 2). It was discovered that active surveillance in Mongolia had found four clusters of H3 (Figure 2A), two clusters of H4 (Figure 2B) and three clusters each of N8 (Figure 2C) and N6 (Figure 2D) genes through phylogenetic analysis. The *HA* and *NA* genes of H3N8 and H4N6 viruses isolated from this study were found to align with the Eurasian lineage, as evidenced by representative clustering. H3 and H4 AIVs were previously shown to be the predominant *HA* subtypes found in the viruses circulating among wild birds in Mongolia [17,18]. In addition, several of these viruses exhibited a close relationship with H3Nx and H4Nx viruses isolated from Korean wild birds during 2017 to 2022 (Figure 2). AIV isolates from Mongolia were found to be closely related to AIV isolates from wild birds in Korea, suggesting that these viruses had been carried by migratory birds between the two countries over 10 years [17]. Intriguingly, among our isolates, A/wild bird/Mongolia/ER30/2021/H3N8 (EPI_ISL_18968158) displayed a low nucleotide identity, below 91.3%, with previously reported N8 genes in GISAID. However, the H3 gene of this virus showed a notably higher nucleotide similarity, exceeding 99.7%, with the H3N8 virus (A/duck/Mongolia/618/2018; EPI_ISL_368635) from 2018. These rare N8 genes may suggest the presence of a minor viral subpopulation infecting wild birds that continues to spread unnoticed.

Phylogenetic analysis of the internal genes revealed that the *PB2*, *PB1*, *PA*, *NP*, *M* and *NS* genes clustered separately from viruses of the Eurasian lineage, including those from Mongolia and Korea. In particular, the PA genes of some H3N8 viruses, including A/wild bird/Mongolia/GU11/2021(H3N8) (EPI_ISL_18968191) isolated in the present study, were highly similar (99.2%) to H5N8 HPAI viruses in Korea in 2021, including A/wild bird/Korea/H379/2020(H5N8) (EPI_ISL_1009706) circulating in Korea in 2021 (Appendix A). In addition, the *PB1* segment of the clade 2.3.4.4b H5N1 HPAI (G1) virus, which has caused global outbreaks since 2020, was also supposed to originate from an A/duck/Mongolia/217/2018(H3N8)-like virus [27]. Therefore, we suggest that some internal genes of H5N8 HPAI viruses, which had affected Korea during the 2020/2021 winter season, may have originated from the H3N8 virus circulating in Mongolia. However, for this type of study, there are still large gaps in known viral sequence data and migratory pathways for the viral introduction between the two countries. Hence, it is imperative to accumulate diverse viruses through such investigations.

Despite the global and unprecedented spread of the Gs/Gd clade 2.3.4.4b H5 HPAI viruses, wild bird surveillance activities in Mongolia since 2021 have failed to detect any HPAI viruses. These results may be attributed to various factors, inadequacies in the existing AI surveillance infrastructure, challenges associated with sample transportation and a notable absence of HPAI viral incursions. However, our genetic analysis of the LPAI viruses isolated in this study consistently highlights striking similarities between viral strains originating from Korea and Mongolia. Such findings strongly suggest an ongoing pattern of seasonal viral exchange facilitated by the migratory behaviors of avian populations between these two regions. Moreover, additional research is warranted to assess the magnitude of migratory birds in the Northern Hemisphere and which bird species predominantly contribute to the transmission of viruses between Mongolia and Korea.

## 4. Conclusions

Mongolia is a highly important habitat in Asia for wild birds, as well as being a destination for wild bird migration. It also hosts substantial populations of wild birds from two major migratory flyways and serves as crucial breeding and pre-migratory staging areas. The findings of the present study suggest that current Mongolian AIV isolates have evolved into genetically diverse genotypes closely resembling those found in wild birds in Korea. Although we could not detect HPAI viruses in this surveillance study, the observation of diverse low-pathogenicity AIVs and some internal genes potentially involved in the emergence of the novel HPAI viruses was very informative for our understanding of the viral situation in Mongolia. It needs to be emphasized that these results need to be interpreted while acknowledging the limited surveillance sensitivity of the fecal samples.

## Figures and Tables

**Figure 1 animals-14-01105-f001:**
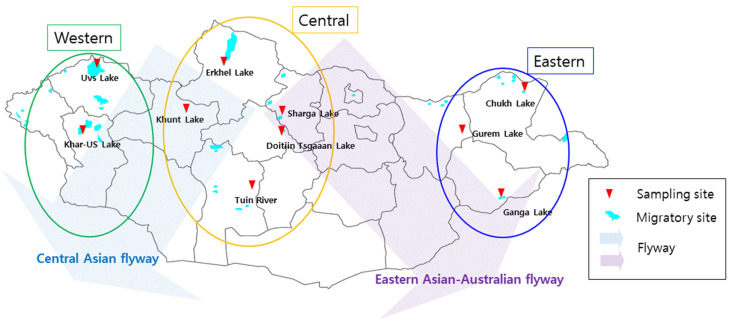
Diagram of the sampling sites and migratory sites of wild birds. In Mongolia, the typical flyways of wild birds include the Central Asian flyway (represented by the transparent blue arrow) and the Eastern Asian-Australian flyway (represented by the transparent purple arrow). Fecal samples of wild birds were collected in three distinct ecogeographic regions of Mongolia: the eastern (blue oval line), central (orange oval line) and western (green oval line) regions. Red triangles indicate sampling sites, and the irregular light blue areas represent the main migratory sites of wild birds.

**Figure 2 animals-14-01105-f002:**
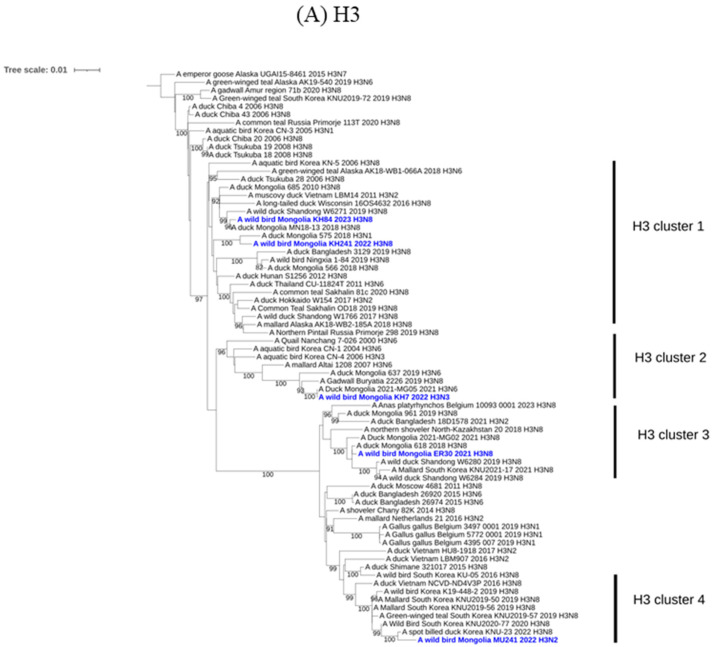
Maximum-likelihood phylogenetic tree for the hemagglutinin (*HA*) and neuraminidase (*NA*) genes. Bootstrap values (1000 replicates) > 70% are displayed at the branch nodes. The Mongolian viruses isolated in this study are highlighted in blue. The scale bar indicates the number of nucleotide substitutions per site. (**A**) Phylogenetic tree for H3 genes. (**B**) Phylogenetic tree for H4 genes. (**C**) Phylogenetic tree for N6 genes. (**D**) Phylogenetic tree for N8 genes.

**Table 1 animals-14-01105-t001:** Numbers of samples tested using real-time RT-PCR and numbers positive for the *M* gene.

	Region of Collection			Total Numbers of Samples	*M*-Positive Samples		Virus Isolation
Year	Western ^(1)^	Central ^(2)^	Eastern ^(3)^		No.	% ^(4)^	
2021	-	2812	1334	4146	39	0.94	30
2022	702	794	641	2137	33	1.54	24
2023	1094	1385	1387	3866	31	0.80	23
Total	1796	4991	3362	10,149	103	1.01	77

^(1)^ The western region: Khar-Us Lake and Uvs Lake. ^(2)^ The central region: Khunt Lake, Erkhel Lake, Sharga Lake, Tuin River and Doitiin Tsgaaan Lake. ^(3)^ The eastern region: Chukh Lake, Gurem Lake and Ganga Lake. ^(4)^ No. of *M*-positive samples/Total No. of samples.

**Table 2 animals-14-01105-t002:** Number of viruses and subtype combinations isolated in the feces of wild birds in Mongolia, 2021–2023.

Subtype	2021	2022	2023	Total
H1N2	0	1	1	2
H2N8	0	2	0	1
H3N1	0	1	0	1
H3N2	0	2	3	5
H3N3	4	2	0	6
H3N8	19	5	6	30
H4N1	0	0	3	7
H4N6	7	4	4	11
H6N1	0	4	2	6
H6N8	0	1	0	1
H10N7	0	2	0	2
H10N8	0	0	3	3
H13N8	0	0	1	1
Total	30	24	23	77

## Data Availability

GISAID: http://platform.epicov.org/epi3/cfrontend accessed on 29 March 2024.

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
