# Peer review of "Surveillance and Genetic Analysis of Low-Pathogenicity Avian Influenza Viruses Isolated from Feces of Wild Birds in Mongolia, 2021 to 2023"

_animals, 2024, doi:10.3390/ani14071105_

Round 1

Reviewer 1 Report

Comments and Suggestions for Authors

The presented study is interesting as it investigates a lage sample size. Overall, the presentation is fine and the study shows where further investigation is needed. But there were sime minor errors, which have to be corrected before publication, especially percentages must be checked again as there were some errors. 

Line 25: 0.99% not 1.09%

Line 70: were selected because they are key wild bird habits
If there was a change in key areas over the years or afterwards, this have to be included in the discussion. Otherwise are would be more suitable.

Line 103: dubble Space "...in Mongolia..."

Line 108: Table 1 is not in the manuscript, but in the supplements. Same goes for Figure 1. Whether indicate that the table is in the supplement or it should be included in the manuscript. In addition, Numbers and percentages of Table 1 must be checked again. The captions of all tables and figures must be self-explanatory and should be supplemented.

Line 112: 1.54% cannot be found in Table 1

Line 119: According to Table 2 H1N2 was only found one time and not twice.

Line 121: Check percentages. 1.09% is not correct.

Line 122: Locational percentage could not be checked. There was no additional table with locations

Line 165: Supplementary Figure 1 is missing in the supplements

Author Response

Response to Reviewer #1 comments

Thank you very much for taking the time to review this manuscript. Please find the detailed responses below and the corresponding revisions highlighted in blue changes in the re-submitted files

The presented study is interesting as it investigates a lage sample size. Overall, the presentation is fine and the study shows where further investigation is needed. But there were sime minor errors, which have to be corrected before publication, especially percentages must be checked again as there were some errors. 

Line 25: 0.99% not 1.09%

- Response :  Thank you for your comment. We apologize for the errors in some of the numbers within Table 1. Upon rechecking the numbers, we have confirmed that the rate is 1.01%, and it has been corrected accordingly. [Line 27, 53]

Line 70: were selected because they are key wild bird habits
If there was a change in key areas over the years or afterwards, this have to be included in the discussion. Otherwise are would be more suitable.

- Response :  Thank you for your comment. We corrected to ‘are’ as your suggestion. [Line 117]

Line 103: dubble Space "...in Mongolia..."

- Response :  Thank you for your comment. We corrected it as you mentioned. [Line 159]

Line 108: Table 1 is not in the manuscript, but in the supplements. Same goes for Figure 1. Whether indicate that the table is in the supplement or it should be included in the manuscript. In addition, Numbers and percentages of Table 1 must be checked again. The captions of all tables and figures must be self-explanatory and should be supplemented.

- Response :  Thank you for your comment. I checked Table and Figure in the manuscript again. I have identified and corrected some error in Table. I have also enhanced the caption of Tables and Figures to make readers more understandable. [Line 115, 164, 199, 206, Line 261~344]

Line 112: 1.54% cannot be found in Table 1

- Response :  Thank you for your comment. We apologize again for the errors in some of the numbers within Table 1. Upon rechecking the numbers, we have confirmed that the rate is 1.54 % not 1.4 %, and it has been corrected it in the table, accordingly. [Line 168 and Table 1]

Line 119: According to Table 2 H1N2 was only found one time and not twice.

- Response :  Thank you for your comment. We corrected to “one each was of subtype H1N2” as you mentioned. [Line 175]

Line 121: Check percentages. 1.09% is not correct.

- Response :  Thank you for your comment. We apologize again for the errors in some of the numbers within Table 1. Upon rechecking the numbers, we have confirmed that the rate is 1.01%, and it has been corrected accordingly. [Line 181]

Line 122: Locational percentage could not be checked. There was no additional table with locations

- Response :  Thank you for your comment. I have included a map in Figure 1 depicting the three region as well as migration route and locational percentage in Supplementary Table S1.

Line 165: Supplementary Figure 1 is missing in the supplements

- Response :  Thank you for your comment. When we submitted our manuscript to journal, we provided all of figures, tables, and supplementary materials. If supplements materials might have been omitted, we will upload supplementary materials to the journal once again.

Reviewer 2 Report

Comments and Suggestions for Authors

Yong-Myung and coworkers performed a surveillance study on avian influenza viruses in Mongolia from 2021 to 2023. Mongolia is an important bird habitat in Asia, which is in the bird flyway in East Asia. In these three years, the authors have collected 10,149 samples, among which 77 strains have been isolated and sequenced. The work in this study is sound. The results are also clearly presented, although some details need to be better described.

Minor problems:

1. Please describe the seasons that most samples are collected in.

2. Since the authors used next generation sequencing, are there any other virus found in the study? Are there any quasispecies of influenza virus in the isolated virus strain? should be discussed.

3. Figure 1 and table 1,2 should be presented in the manuscript, they are not supplementary materials. The figure 1a, b, c, d, should be clearly cited in the manuscript.

4. A brief map should be included showing the lakes in Mongolia and the birds’ flyway.

Comments on the Quality of English Language

English is find. Minorediting is required.

Author Response

Response to Reviewer #2 comments

Thank you very much for taking the time to review this manuscript. Please find the detailed responses below and the corresponding revisions highlighted in blue changes in the re-submitted files

Yong-Myung and coworkers performed a surveillance study on avian influenza viruses in Mongolia from 2021 to 2023. Mongolia is an important bird habitat in Asia, which is in the bird flyway in East Asia. In these three years, the authors have collected 10,149 samples, among which 77 strains have been isolated and sequenced. The work in this study is sound. The results are also clearly presented, although some details need to be better described.

Minor problems:

  1. Please describe the seasons that most samples are collected in.

- Response :  Thank you for your comment. The migratory birds moving to Korea breed in northern areas such as Mongolia during the summer season. However, during the winter season, the temperatures drop too low, causing the birds not to inhabit the region, making it difficult to collect samples. Therefore, the samples were collected evenly from April to October (Spring, Summer and Autumn).

  1. Since the authors used next generation sequencing, are there any other virus found in the study? Are there any quasispecies of influenza virus in the isolated virus strain? should be discussed.

- Response :  Thank you for your comment. In this study, We targeted the genetic segments of the AI virus for amplification and conducted NGS. As a result, we were unable to detect any quasispecies other than AI. Quasispecies of other AI was present in the result of NGS in some samples, but sufficient gene sequences were not obtained for genetic analysis. Although some samples were suspected to be mixed, only those with clearly confirmed genetic sequences were analyzed in this study. We described this in the manuscript. [Line 175]

  1. Figure 1 and table 1,2 should be presented in the manuscript, they are not supplementary materials. The figure 1a, b, c, d, should be clearly cited in the manuscript.

- Response :  Thank you for your comment. We presented Figure1, Figure 2a,b,c,d and Table 1,2 as you mentioned. [Line 115, 164, 199, 206]

  1. A brief map should be included showing the lakes in Mongolia and the birds’ flyway.

- Response :  Thank you for your comment. We add additional figure presenting brief map as Review 1 suggestion, too. [Figure 1] 

Reviewer 3 Report

Comments and Suggestions for Authors

A large number ( 10,149) fecal samples were collected from wild birds in western, central and eastern Mongolia from 2021 to 2023 for being tested for influenza A. Various subytpes were detected and sequenced.

I enjoyed reading the manuscript and for sure it presents valuable data, however some additional information should be added.

1.        The three geographical regions (Western, Central, Eastern) should be better described, please add coordinates and a map would be recommended. For the map it would be interesting to add migratory routs and main migratory sites.

2.         What kind of birds are found there? Aquatic? Migratory? Is it not possible to evaluate from what species was faces?

3.        What do you mean by fresh faecal samples? They were collected in the moment of defecation? Did you have same methodology of collection, like each 20m in distance? Alonge some trail?

4.        How the faces were collected? With some swabs?

5.        How the faces were preserved till get to the laboratory?

Author Response

Response to Reviewer #3 comments

Thank you very much for taking the time to review this manuscript. Please find the detailed responses below and the corresponding revisions highlighted in blue changes in the re-submitted files

A large number ( 10,149) fecal samples were collected from wild birds in western, central and eastern Mongolia from 2021 to 2023 for being tested for influenza A. Various subytpes were detected and sequenced.

I enjoyed reading the manuscript and for sure it presents valuable data, however some additional information should be added.

  1. The three geographical regions (Western, Central, Eastern) should be better described, please add coordinates and a map would be recommended. For the map it would be interesting to add migratory routs and main migratory sites.

- Response :  Thank you for your comment. I have included a map in Figure 1 depicting the three region as well as migration route and migratory sites as Reviewer 1 and 2 suggestion, too.

  1. What kind of birds are found there? Aquatic? Migratory? Is it not possible to evaluate from what species was faces?

- Response :  Thank you for your comment. I understand ‘faces’ as ‘feces’. When collection fecal samples, the species and numbers of wild birds inhabiting the area at that time were also surveyed. We primarily collected samples around lakes and rivers, and the wild birds species inhabiting the area included herons, geese, gulls, and wild ducks that were aquatic and migratory birds. Observation, looking through binoculars and telescopes were used to define the targeted migratory waterfowl (Anseriformes) bird species include ducks, geese, and swans and population numbers. The differentiation of avian species from feces can be inferred through the genetic sequencing of mitochondrial DNA within the feces but it is not easy to make precise determination. Although this study did not prioritize virus gene analysis, it seems that further research would be necessary.

  1. What do you mean by fresh faecal samples? They were collected in the moment of defecation? Did you have same methodology of collection, like each 20m in distance? Alonge some trail?

- Response :  Thank you for your comment. Fresh fecal samples were collected from locations where waterfowl were observed congregating or roosting as described in the manuscript [Line . Fecal samples collected after observing defecation. Droppings that appeared freshly passed (within the half hour), as judged by moisture, size, shape, and appearance of their surface, were sampled. We did not focus on specific distance or trails, but fecal samples were collected within 20 meters of the shoreline where a flock of birds was gathered.

  1.  How the faces were collected? With some swabs?

- Response :  Thank you for your comment. The fecal samples were collected directly and From each sample, small part of feces (5-8 g) was conveyed into 15 ml conical tube or 5.5*8cm zipper bag. The samples were transported promptly to the laboratory for test.

  1. How the faces were preserved till get to the laboratory?

- Response :  Thank you for your comment. The collected fecal samples were stored in a small portable refrigerator and transported to the laboratory within 24 hours under refrigerated conditions (4~8 degrees Celius), as described additionally in the manuscript.[Line 118]

Round 2

Reviewer 3 Report

Comments and Suggestions for Authors

I do not have more comments. Congrats!